Environment and density-dependency explain the fine-scale aggregation of tree recruits before and after thinning in a mixed forest of Southern Europe

Rodríguez-Pérez Javier javier.rodriguezperez@unavarra.es 1 2
Imbert Bosco 1 2
Peralta Javier 2
1 Institute for Multidisciplinary Research in Applied Biology (IMAB), Centro Jerónimo de Ayanz, Universidad Pública de Navarra , Pamplona , Navarra , Spain
2 Department of Sciences, Campus Arrosadía, Universidad Pública de Navarra , Pamplona , Navarra , Spain
Wehenkel Christian
Electronic publication date: 2022 Sep 12
Publication date: 2022
Volume: 10
Electronic Location ID: e13892
Received 2022 Jan 20; Accepted 2022 Jul 22
Copyright: ©2022 Rodríguez-Pérez et al.
Copyright year: 2022
Copyright holder: Rodríguez-Pérez et al.
License: This is an open access article distributed under the terms of the Creative Commons Attribution License, which permits unrestricted use, distribution, reproduction and adaptation in any medium and for any purpose provided that it is properly attributed. For attribution, the original author(s), title, publication source (PeerJ) and either DOI or URL of the article must be cited.
License URL: https://creativecommons.org/licenses/by/4.0/

Keywords: Mixed-forests, Point-pattern analysis, Tree-recruit competition, Local scales, Biotic interactions

Funding: The Spanish Ministry of Economy and Competitiveness AGL2006-08288 AGL2009-11287 “La Caixa” and “Caja Navarra” Foundation, in the framework of UPNA’s “Captación de Talento” program LCF/PR/PR13/51080004 This research was funded by the Spanish Ministry of Economy and Competitiveness (AGL2006-08288 and AGL2009-11287). Javier Rodríguez-Pérez was funded from the “la Caixa” and “Caja Navarra” Foundation, under agreement LCF/PR/PR13/51080004 in the framework of UPNA’s “Captación de Talento” program. The funders had no role in study design, data collection and analysis, decision to publish, or preparation of the manuscript.

==============================
Thinning in forest management primarily reduces the density of trees and alters the patchiness and spatial complexity of environmental factors and individual interactions between plant recruits. At fine spatial scales, little is known about the relative weight of ecological processes affecting tree regeneration before and after thinning events. Here we studied the density and aggregation of tree recruits in fully-mapped plots located in mixed forests in Northern Iberian Peninsula (Southern Europe) for over four years, which comprises one year before and three years after a thinning event. We applied spatial point-pattern analyses to examine (a) the aggregation of recruits, and their association with trees and (b) the relative effect of both environmental (i.e., the patchiness of the local environment) and density-dependent factors (i.e., the aggregation of trees and/or recruits) to predict the density, aggregation, and survival of recruits. We found, in thinning plots, that recruits were less dense, their aggregation pattern was more heterogeneous, were distributed randomly in respect of trees and their survival was almost unaffected by the tree proximity. By contrast, recruits in control plots were denser, were only aggregated at distances lower than 1.0 m, were closer to trees, and such closer distance to trees affected negatively in their survival. Independently of the treatment, the aggregation of recruits was chiefly determined by the density-dependent factors at less than 1.0 m and environmental factors at distances beyond that proximity. Overall, our results suggest that thinning affected the aggregation of recruits at two spatial scales: (a) by favoring the tree-recruit and recruit-recruit facilitation at less than 1.0 m and (b) by modifying spatial heterogeneity of the environment at distances beyond that proximity.

Introduction

Spatial patterns of species or communities reflect underlying ecological processes which are occurring or have occurred in past, including historical, environmental heterogeneity and/or biotic processes (McIntire & Fajardo, 2009). Disturbances, both natural or human-induced, alter the quantity, quality, spatial arrangement, and patchiness of local environments, and leave a lasting imprint on the density and/or aggregation of species communities. In forests, disturbances affect the spatial heterogeneity of microclimate (e.g., Ma et al., 2010; Primicia et al., 2013), leading to creating new conditions with potential impacts on the abundance and composition of plant communities (Frelich, 2002). For instance, disturbances have effects on the fine-scale environment which create new opportunities for colonization and regeneration of tree species. Likewise, biotic interactions are key in explaining plant coexistence at local scales (Thébault & Fontaine, 2010; Walther, 2010), leading to generating spatial gradients of competition to facilitation dependent on the fine-scale environment (Tilman & Kareiva, 2018). In forests, disturbances alleviate competition between neighboring individual plants by reducing their density and aggregation and that may generate quick and subsequent changes in early regeneration stages (e.g., Denslow, 1995; Bailey & Tappeiner, 1998; Dolanc, Gorchov & Cornejo, 2003; Yeo & Lee, 2006). For all the above reasons, understanding the interaction of the local neighborhood with coarser-scale processes generated by disturbances, and their spatial and temporal scales of interest, is undoubtedly a critical issue in forest dynamics research (Gratzer et al., 2004).

Thinning is an important human-based disturbance and a widespread practice to manage forests, with strong implications for carbon cycle, microclimatic conditions, tree ecophysiology, and tree population dynamics (Oliver et al., 1996). Thinning consists of selective removal of a proportion of trees, and this action has primary effects on the patchiness and spatial complexity of fine-scale environmental factors. By reducing tree density and increasing their average size, temperature, light conditions, canopy composition, and stand structure are generally altered by thinning (e.g., Battles et al., 2008; Ganey & Vojta, 2011), whereas the removal of canopy and/or understory layers leads to a modification in environmental factors affecting forest aggregation and patchiness (Barbier, Gosselin & Balandier, 2008). Likewise, thinning impacts on the short-to-medium term of the tree regeneration stages as a consequence of alteration of fine-scale environmental patterns and the local interactions between individual trees, notably by decreasing aggregation of trees and/or plant recruits (i.e., density-dependent factors) and by alleviating the tree-recruit competition for resources. In addition, stand-level environmental variability may interact with density-dependent patterns, generating as a result changes in the aggregation of recruits after thinning events (Wild et al., 2014). Factoring out the effect of environmental from the density-dependent factors on tree regeneration has been largely unexplored from the fine-scale spatial perspective (Perry, Enright & Lamont, 2008; Shen et al., 2013; Pescador et al., 2020), which is key to providing more efficient tools to maintain long-term resources in forest systems.

Here, we studied the effect of thinning on spatial patterns and tree regeneration. We focus on a site in southern Europe, located in a transition area between Mediterranean and temperate climates, which include tree species typical in both biogeographic regions. To do so, we studied the aggregation of recruits in six fully-mapped plots, which comprise three thinning and three non-thinning plots (control, hereafter). The spatial patterns of recruits are probably among those with a more evident spatial signal as they can quickly respond to underlying environmental and density-dependent factors (Moeur, 1997; Kenkel, 1988; Plotkin, Chave & Ashton, 2002). We studied all six plots one year before and three years after a thinning event: with the latter sampling design, we were able to discriminate between historical processes derived from the local dynamics and thinning events. To overcome such objectives, we applied Spatial Point-Pattern Analyses (SPPA) to derive specific point-process models which represent spatially-explicit biologically meaningful hypotheses. SPPAs could provide a causal relationship between ecological processes and the spatial patterns derived from observational data, such as the associations between the point patterns of ecological objects or the relationship between the distribution of resources and former point patterns (Wiegand & Moloney, 2014). We performed SPPAs of independent plots as we expected plot-level variability derived from the local environmental conditions. Likewise, we also expected common processes derived from each treatment affecting the aggregation of recruits, and we thus pooled results into control vs. thinning plots. We first assessed if the density, aggregation, and survival of recruits is influenced by the proximity of trees and recruits. We hypothesized that thinning potentially alleviates the competition between adult trees and recruits as a consequence of partial tree removal. Secondly, we tested if recruits were influenced by the within-plot spatial patterns defined by (i) the environmental and/or (ii) the density-dependent factors of trees and recruits, to weight their relative effect of both spatial patterns on predicting the density, aggregation, and survival of recruits before and after thinning. We hypothesized that the spatial patterns of recruits in thinned plots depended more on fine-scale environmental heterogeneity as thinning alleviates density-dependent factors.

Methods

Study area

The study site is a forested area located in Aspurz (Navarre, Spain), in a north-facing oriented foothill of mounts of the western pre-Pyrenees at 590–630 m a.s.l. (Appendix S1; Fig. S1). This area is in the fringe of the mid-European and Mediterranean biogeographical regions, with a sub-Mediterranean climate transitional between temperate and Mediterranean. There is a mild water deficit during summer, and frequent frosts from winter to early spring. In the last 20 years, mean annual precipitation was 937.2 mm and mean annual temperature 12.0 °C; latter data from Navascués, a weather station located at 2.7 km north and 615 m a.s.l. During summer, cumulative rainfall were 174, 188.7, 284.4 and 157.3 mm and mean maximum temperatures were 27.1, 29.4, 26.9 and 27.3 oC for 2008, 2009, 2010 and 2011, respectively.

The study site’s vegetation is composed by a mixed forest of Pinus sylvestris L. (Scots pine), capitalizing most canopy that shares with broad-leaf species. In our study’s site, forest landscape is composed by extensive secondary pine woodlands, patches of Fagus sylvatica L. (European beech) woods located in northward slopes with favored water balance and of Quercus pubescens Mill. (Pubescent oak) woodlands elsewhere. Quercus ilex L. (Holm oak), a typical Mediterranean tree, dominates woodlands in the driest biotopes, as rocky limestone outcrops. Additional tree species occur in our study site, but at very low densities.

Since the beginning of the 20th century, Scots pine has been promoted in our study’s area through the replacement of broad-leaf species usually displaced by recurrently forestry management, leading to generate landscapes dominated by secondary pine forests (Costa, Morla & Sainz, 2005). Currently, many beech trees are co-dominant or dominant, and occupy, on average, 39.4% of canopy (Arozaena-González, 2020). Q. pubescens and Q. ilex, occurring in the study area, have emerged through a natural and sexual regeneration. Currently, the majority of individuals of Q. pubescens and Q. ilex are present as early stages of regeneration, with few individuals arriving to the mid-stages of canopy strata (7.3% and 0.5% of the mixed canopy, respectively; Arozaena-González, 2020).

Experimental design and tree and recruit samplings

In 1999, the Servicio Forestal del Gobierno de Navarra set up nine plots of 30 × 40 m or 0.12 ha in size in our study site which accounted of similar environmental conditions (Appendix S1, Fig. S1). In 1999 and 2009, two thinning intensities over Scots pine trees were performed (20% and 30% (1999) or 40% (2009) of basal-area removed) and one control treatment (no thinning, 0%), which resulted in three plots for each treatment. In our study we focused on the detailed analysis of three 0% or control plots and three 30/40% or thinned plots, as there is detailed information of the tree and recruit mapping and the fine-scale distribution of environmental variables; see Appendix S1 for detailed information of the data available for each plot. We thank the Government of Navarra for the experimental setting of silvicultiral treatments, and the Concejo de Aspurz for permitting access to the study plots.

In 2009, we surveyed the position and size of trees right before the second thinning event in control and thinned plots. We performed a single survey due to the fact that most individuals measured more than 12.0 m, with very low chances of mortality during the study-time period. For each plot, we mapped the relative position of each individual tree (height >2.0 m) and we took DBH measures of their size. For details, see Appendix S1 (Fig. S2).

From 2008 to 2011, we surveyed recruits including one year before and three years after thinning. We defined as recruit based on a size threshold of tree recruits species and individuals <200 cm. We focused on recruits of F. sylvatica, Q. pubescens and Q. ilex for several reasons. First, broad-leaf species are dominant in the early stages of regeneration at our study site, with very low natural regeneration of P. sylvestris recruits. In particular, we observed a peak of seed production and germination of P. sylvestris in 2009, but, after three years, less than 5% of its recruits remained alive (García-Sancet, 2017); the latter observation suggests that P. sylvestris recruits have probably little impacts on the density, aggregation and survival of the rest of recruits. Second, most broad-leaf trees within study plots have not arrived at the upper part of the forest canopy and still do not produce seeds. In our study plots, most reproductive trees belong to P. sylvestris. F. sylvatica is the exception, and it represents the 4.42% ± 0.585 (mean ± s.e.) of the largest (DBH >10 cm) individuals of each plot. However, we can not discard that other broad-leaf mature trees were located outside our study plots, providing seeds and seedlings. Overall, we were unable to calculate the contribution of the limitation of seed dispersal of broad-leaf recruits but, if that existed, it was likely low. For each plot and year, we mapped individual recruits in a regular grid sampling of 300 2.0 × 2.0 m cells defined by 15 columns and 20 rows. For each recruit, we mapped its relative position within each cell, its size (i.e., height to the nearest mm) and its survival (i.e., alive vs dead); over the study years, recruits growing >2.0 m were tagged as trees and their height was not longer measured. For details, see Appendix S1 (Fig. S2).

In order to describe the year and treatment variability at our study site, we first looked for the density of individual trees at each plot. For each year and plot, we calculated the averaged density of trees and recruits, averaged for each treatment (mean ± standard deviation).

Maps of environmental variables

Recruit’s survival is strongly affected by resource’s heterogeneity, and we thus constructed a set of maps or spatially-explicit covariates which include the within-plot environmental heterogeneity. In 2008, we obtained the canopy richness by means of a field survey leading to measure areas of vertical projection of tree crowns beyond 2.0 m in height over the forest ground. For each plot, we accurately mapped the canopy area comprised by homogeneous areas of a single species and of at least one individual tree. The species classification of tree species was based on foliage morphology. Only canopy of broad-leaf species was mapped, as to the canopy of P. sylvestris trees showed a relatively homogeneous distribution within plots. For each plot, we calculated maps of the species richness of the canopy by summing individual maps (Appendix S1; Fig. S3).

In 2004, we measured light variability when passing through canopy using hemispherical pictures (HP). In each plot HPs were sampled in a regular grid of 300 cells, resulting in at least 15 HP per plot; additional HP pictures were obtained in areas with with high vegetation heterogeneity. Each HP was acquired at 0.5 m height and incorporates a compass providing the Magnetic North. We additionally acquired metadata of the geographical location, date and atmospheric parameters which depend on our study site. Each HP was taken under good meteorological conditions, assuring stable and similar sky conditions. HPs were retaken if reflections on trees of excess light was observed, due to the open sky could be overestimated. For each HP, we computed the canopy openness, Leaf Area Index (LAI), the averaged direct and diffuse light (both for the whole year and for the summer period) and the number and the median and maximum time duration of sunflecks. For each plot, we computed maps of continuous covariates of the above-defined light indexes (Appendix S1; Fig. S4).

In 2008, we measured the variability of the forest understory bellow 2.0 in height. We focused on the most common understory species in our study site, namely Hedera helix L. (ivy), Pteridium aquilinum (L.) Kuhn (common bracken), Rubus ulmifolius Schott (bramble) and the moss Scleropodium purum (Hedw.) M. Fleisch (Neat Feather-moss); latter species, where present, monopolize the forest ground due to their clonal or vegetative growth forming dense masses of understory vegetation. We also considered the percentage of area covered by F. sylvatica representing branches of trees projected bellow 2.0 m. In each plot, we measured sampling points from a regular grid of 300 cells. In each sampling point, we measured the percentage of area covered by understory vegetation defined as the percentage of vertical projection area bellow 2.0 m in height, in respect to the total area. For each plot, we computed maps of covariates of the proportion area covered by each species at the forest understory (Appendix S1; Fig. S5).

Due to the fact that covariates derived from different data sources and surveys, we created maps sharing a common grain or resolution capturing the fine-scale variability of environmental covariates. For covariates sampled at regular grids of 300 2.0 × 2.0 cells (i.e., light variables, forest understory), we computed maps of covariates at 1.0 × 1.0 m cells using techniques of geospatial statistics. Specifically, we computed a ordinary kriging fitting a spherical variogram model in statistical language R (R Core Team, 2020) using the library gstat (Gräler, Pebesma & Heuvelink, 2016). Maps are shown in Appendix S1.

Generalities of the point-pattern analysis

We used techniques of spatial point-pattern analysis (SPPA) (Wiegand & Moloney, 2004; Law et al., 2009) to describe the aggregation of trees and recruits and the association of environmental (i.e., the patchiness of the local environment) and density-dependent factors (i.e., the local aggregation of trees and/or recruits) explaining the aggregation of recruits within plots. To describe the statistical properties of the observed aggregation of recruits (Diggle, Ribeiro & Christensen, 2003; Illian et al., 2008), we used different summary statistics (see Appendix S1) aimed to summarize the observed point pattern. The observed point patterns of recruits were contrasted to spatially-explicit null models derived from multiple (199) realizations. For each hypothesis, we implemented specific point-process models providing clues of the ecological hypotheses or spatial-explicit processes likely conditioning our study system (Wiegand & Moloney, 2004; Jacquemyn et al., 2007); see Table 1 for the summary statistics applied for each specific test. We used summary statistics to summarize the observed point pattern and the modeled point process. In particular, we based our analyses on the pair-correlation function g(r), which calculates the density of points (i.e., recruits or trees) at distances from a typical point of the pattern; the latter function use rings as the observation window to calculate the density of points and it is thus non-cumulative. In addition, we performed additional analyses based on the L-function L(r), which uses disks as observation window; the L(r) function is the accumulative version of the g(r) function and calculates the density of points (i.e., recruits or trees) within the distance of a typical point of the pattern. To describe additional properties of the spatial patterns, we used the bivariate g(r) and L(r) distribution functions, which gives the fraction of points of type 1 that have nearest points of type 2 at and within, respectively, increasing distances (Diggle, Ribeiro & Christensen, 2003; Illian et al., 2008). See Appendix S1 for details of summary statistics.

Table 1 Details of specific tests and hypotheses, summary statistics and null models.

Hypotheses	Summary statistics	Null models	
1) Analysis 1: Description of aggregation of trees and recruits	
	
i) Aggregation of recruits and their association between them	Univariate g(r)	Univariate homogeneous Poisson process: Recruits are randomly and independently distributed.	
ii) Relationship between trees and recruits	Biariate g(r)	Independent “Qualitative” marks with null model of “antecedent conditions”: Trees are held fixed, whereas recruits were randomly and independently distributed.	
iii) The survival of recruits based on the tree location	Bivariate Lij (r)	Null model of “independent labeling” or “trivariate random labeling”: the survival of alive or dead recruits is conditioned on tree locations.	
2) Analysis 2: Factors explaining the density and aggregation of recruits	
	
iv) Environmental factors explain the aggregation of recruits	Univariate g(r)	Inhomogeneous Point Process: The intensity function λ(x) of the point process in the recruit location and depends on the environmental covariates.	
v) Density-dependent factors explain the aggregation of recruits	Univariate g(r)	Inhomogeneous Point Process: The intensity function λ(x) of the point process in the recruit location and depends on the density-dependent covariates.	

From latter realizations, we estimated the envelopes encircling the 95% range of values of the summary statistic under a given point process. Departures from the specific point process occurred if the observed point pattern (described by summary statistics) fell outside the simulation envelopes generated by the point process (Wiegand & Moloney, 2004). Slight departures of observed patterns from simulation envelopes were not interpreted as being meaningful in order to accept false null hypotheses (e.g., Wiegand & Moloney, 2014). Thus, we performed goodness-of-fit tests by providing a single statistic along all distances r, ui, which shows the total squared deviation between observed and simulated patterns (see Baddeley, Rubak & Turner, 2015; Loosmore & Ford, 2006).

All SPPAs were performed in statistical language R (R Core Team, 2020) using the libraries spatstat.core (Baddeley, Rubak & Turner, 2015) and ecespa (De la Cruz, 2008). We restricted our analyses to the rectangular area limiting our sampled plots (see Appendix S1). For each plot we performed independent analyses derived from patch dynamics of our study site. Because we expected “a priori” common ecological processes occurring between control and thinned plots, we additionally combined the results of the individual analyses using techniques for replicated patterns (i.e., Law et al., 2009). Latter techniques allow for a combination of all summary statistics derived from independent samples and analyses into a single “master” statistical test. Such consensus statistical test represents the average of the summary statistics of independent analyses, weighted by the number of focal points in the focal pattern (Illian et al., 2008). These analyses allowed us to focus on the average processes, rather than on the potential variability of each plot. The estimation of summary statistics requires avoiding or correcting for the bias produced by being near the edge of the observation window. In our particular case, those points located in the boundaries of our study plots will generate incomplete rings or disks and thus biased estimates of summary statistics. To solve this issue, we performed edge corrections to minimize the biases by the edge effects. To do so, we applied different methodologies of edge corrections which calculate unbiased estimates of summary statistics (Baddeley, Rubak & Turner, 2015).

Analyses of spatial point patterns

We performed analyses aimed at (1) describing the aggregation of trees and recruits and (2) detecting the factors explaining the density and aggregation of recruits. For the first objective, we specifically (i) analyze the aggregation of recruits and their association between them, (ii) the relationship between trees and recruits and (iii) the survival of recruits and its dependence on the tree location. For the second objective, we aimed at disentangling the relative importance of (iv) environment and/or (v) density-dependent factors explaining the aggregation of recruits. A summary of the specific tests and hypotheses, summary statistics and null models are in Table 1.

Analysis 1: Description of aggregation of trees and recruits

To analyze the aggregation of recruits and their association between them, we carried out “unmarked” PPAs (Diggle, Ribeiro & Christensen, 2003; Illian et al., 2008). For each plot, we evaluated the aggregation pattern of recruits against the simplest null model of a point process, the Homogeneous Poisson Process (HPP). Each realization of the HPPs follows a Complete Spatial Randomness (CSR) generated by randomly and independently drawing points from the limits of our study plots. To do so, we computed the univariate g(r) recruits with Ripley’s isotropic correction (Baddeley, Rubak & Turner, 2015).

For analyzing the relationship between trees and recruits, we carried out analyses based on independent “qualitative marks” (Diggle, Ribeiro & Christensen, 2003). As a null process or model, we used tests of “antecedent conditions” as we assumed that trees may influence the location of recruits, but not the other way round (Velazquez et al., 2015). Specifically, the spatial location of the observed trees were held fixed, whereas the spatial location of recruits were randomly and independently distributed. As a summary statistic, we computed the g(r) function with the Ripley’s isotropic correction (Baddeley, Rubak & Turner, 2015).

To analyze if the survival of recruits depends on the tree location, we carried out analyses based on “independent labeling” (De la Cruz et al., 2008) or “trivariate random labeling” (Wiegand & Moloney, 2014). Specifically, the null process or model assumes that the survival of recruits are independent to the location of trees and that trees influences the survival of recruits. As a null model, the survival (i.e., alive vs dead) of recruits is randomly permuted while keeping the locations of trees fixed. Departures from the confidence intervals of the null model indicates that the process generating the recruit survival is conditioned on the locations of trees. As a summary statistic, we used the bivariate L(r) with the Ripley’s isotropic corrections (Baddeley, Rubak & Turner, 2015). Assuming a complete random labeling, L(r) >0 indicates that, at a distance r, alive or dead recruits are more aggregated with trees than expected under random labeling.

Analysis 2: Factors explaining the density and aggregation of recruits

The relative importance of environment and/or density-dependent factors explaining the aggregation of recruits was assessed by measuring the effect of within-plot covariates in a location of recruits (Baddeley, Rubak & Turner, 2015). Specifically, we fitted Inhomogeneous Poisson Processes (IPP) as we expected the existence of uneven distribution of fine-scale ecological factors, mediated the uneven distribution of covariates, conditioning the prediction of recruits in each location.

For environmental factors, we included 12 maps of covariates with uneven variability of their values in each plot. Namely, the species richness of the canopy (Canopy), the canopy openness (CanOpen), leaf area index (LAI), the average diffuse light, both for the whole year (DiffBelow.Yr) and for the summer period (Diff.Below), the number, median duration (s) and maximum duration (s) of sunflecks (N.Sunflecks, Mdn.Sunflecks, Max.Sunflecks, respectively), the relative abundance of F. sylvatica (Fs), H. helix (Hed), R. ulmifolius (Rub), P. aquilinum (Pter) and S. purum (Scl). All of those maps of covariates did not vary along years due to sampling period occurred in a single year (Appendix S1; Figs. S3 to S5).

In addition we found the existence of strong within-plot inhomogeneity in the distribution of recruits (see Results), which suggests that the aggregation of trees and/or recruits could affect the recruit’s presence in certain areas of the plot. We included five maps of density-dependent covariates, calculated from intensity function λ maps of trees and recruits. Namely, the density of trees (Dens.adult), the density of the size of recruits (Dens.size.rec), and the density of richness (Dens.rich.rec) and the Shannon index of recruits (Dens.shan.rec). Except for the density of trees (sampling in 2009), the density-dependent covariates and maps were calculated for each year (Appendix S1; Fig. S6).

For each year and plot, we fitted two sets of IPP models, assuming that the location of each recruit either depends on (i) environmental or (ii) density-dependent covariates. Previous to fit both sets of models, we performed pair-wise cross-correlation analysis to discard (Pearson r > 0.7) highly correlated variables (Appendix S1; Fig. S7). Each set of models depend on a combination environmental covariates (i.e., Canopy, CanOpen, LAI, DiffBelow, Fs, Hed, Rub, Scl) or density-dependent covariates (i.e., Dens.adult, Dens.size.rec, Dens.rich.rec, Dens.shan.rec) with low cross-correlation between them (Appendix S1; Fig. S7). For each model, we performed a stepwise model selection based on removing, in each step, a single covariable based on minimizing the AIC score (Baddeley, Rubak & Turner, 2015); the latter approach was applied recursively until no more terms could be deleted or AIC increases. For each year and plot, we obtained estimates of the predicted probability maps, and we extracted the value of the predicted probability for each recruit location, representing a range of continuous values of optimal conditions for factors: larger values of predicted values in each recruit location suggest that they had better conditions for either environmental or density-dependent maps or covariates.

Results

Description of the density of trees and recruits

For each year we recorded, on average, 1367 ± 409 trees and 1218 ± 703 recruits in all six plots (mean ± standard error across years). Comparing species, the most abundant trees were P. sylvestris (956 ± 334), followed by F. sylvatica (176 ± 60.5), Q. pubescens (137 ± 120) and Q. ilex (98.5 ± 107), whereas recruits were dominated by Q. pubescens (642 ± 361), followed by Q. ilex (478 ± 278) and F. sylvatica (97.6 ± 80.7).

We found that trees were denser in control than in thinned plots in the year before thinning (2008), but that difference disappeared after thinning (Fig. 1A). Latter pattern were mimicked by recruits, which were denser in thinned plots before the thinning event and arrived at a range of comparable values to that of control plots over the studied years (Fig. 1B). Before the thinning event, recruit’s survival was independent of the treatment, but we found a steadily and increasing higher survival in thinned than control plots after thinning (Fig. 1C). We additionally found a proportionally higher between-plot variability in control plots for density of trees and, notably, of recruits. However, between-plot variability was higher in thinned plots for the proportion of recruit’s survival (Fig. 1C); the latter spatial patterns were consistent over the studied years.

Figure 1 Density of trees and recruits and survival of recruits for control and thinned plots.

For each treatment, bars represent the average and their associated confidence intervals. The density of trees and recruits were calculated at square-meters, whereas recruit’s survival was calculated for the whole plot. For each variable and year, significance between treatments were showed over bars (i.e., Tukey tests including treatment and year as independent factors). Probabilities: *** p < 0.001, * p < 0.05 and n.s. p ≥ 0.05.

Description of aggregation of trees and recruits

Overall we found that the spatial aggregation of recruits departed from complete spatial randomness (CSR), arriving up to ten times the average density of recruits at less than 0.5 m; the latter pattern was consistent over the studied years (Fig. 2). Pooling control plots we found consistent aggregations at less than 0.5 m, including small but inconsistent departures from CSR at 2.0 m (Fig. 2). In thinned plots, by contrast, we found aggregation at several distances, notably at less than 0.5 m and from 1.5 to 2.25 m and beyond; the latter pattern was consistent over the studied years (Fig. 2). Comparing the results of independent plots, we found that the majority of plots had positive departures from CSR at several distances, typically at less than 1.0 m and beyond; the latter patterns were consistent over the studied years but strongly varied between plots (Appendix S2; Fig. S1).

Figure 2 Analyses of the spatial aggregation of recruits for control (left panels) and thinned plots (right panels).

For each treatment, the results of each of the three plots are averaged. The pair correlation functions g(r) from the data (black lines) are contrasted to simulation envelopes (grey area) derived from the 199 realizations (i. e., including the 5th lowest and highest values) of the fitted complete spatial randomness (CSR) null models. The expectation of random patterns appear as dashed line. In upper plots we show the spatial correlogram of the 2008 year, whereas in lower plots a simplified representation of yearly spatial correlograms is shown. In lower panels values above simulation envelopes are in green, values below confidence envelopes are in red and values within the envelopes (random) are in white.

We tested for differences in the spatial association of trees and recruits and found that recruits in control plots were closer to trees at less than 2.5 m, whereas in thinned plots the location of recruits was independent of trees, with small but significant departures up to 6.0 m (Appendix S2; Fig. S2). Comparing independent plots, we found consistent aggregation of recruits to trees in plot 3 and small departures of aggregation in plots 2, 5, 7 and 9, in all them up to 6.0 m. By contrast, in plot 4 (control), we found repulsion of recruits to trees from 1.5 to 6.0 m (Appendix S2; Fig. S3).

We additionally found that the recruit’s survival depended on the tree distance. In control plots, recruits had higher survival than expected by random beyond 2.0 m; In thinned plots recruits had higher survival beyond 6.0 m from trees (Appendix S2; Fig. S4). Comparing independent plots, we did not found a consistent evidence of higher recruit survival as a function of the tree distance in control plots 3 and 9 from 2.0 to 8.0 m,and thinned plots 2 and 7 beyond 6.0 m. In control plot 4 by contrast, we found that recruits had lower survival than expected by random if trees were beyond 1.5 m from trees whereas in plot 5 we only detected higher survival of recruits at around 2.0 m; the latter pattern was consistent over the studied years (Appendix S2; Fig. S5).

Factors explaining the density and aggregation of recruits

Overall we found that the factors explaining the density and aggregation of recruits were consistent over the studied years. Specifically, the recruits were denser in richer tree canopies, in areas with higher proportion of P. aquilinum, and in thinned plots (Appendix S2; Table S1). Comparing independent plots, the canopy richness had effects on plots 4, 5 and 9, LAI in thinning plots 5 and 7, DiffBelow on plots 2, 7 and 9, thinned plots 4 and 7, the proportion of F. sylvatica in control plot 9, the proportion of R. ulmifolius on plots 7 and 9, and the proportion of P. aquilinum on plot 3, 7 and 9 (Appendix S2; Table S3).

Regarding the density-dependent factors, recruits were denser in areas with lower tree density (i.e., but only in 2011), in areas with larger recruits, in areas with higher species richness and in areas with lower values of the Shannon index (Appendix S2; Table S2); in the case of the treatment, recruits were denser in thinned plots, but only in 2009. Comparing independent plots, the density of trees affected on control plot 3, the size of recruits affected positively in all plots, the richness of recruits affected positively on plots 2, 4, 5, 7 and 9 and the Shannon index affected negatively all plots except the control plot 3 (Appendix S2; Table S4).

Figure 3 Analyses of the spatial aggregation of recruits affected by environmental factors for control (left panels) and thinned plots (right panels).

The pair correlation functions g(r) from the data (black lines) are contrasted to simulation envelopes (gray area) based on maps of environmental covariates of each plot. For further conventions, see Fig. 2.

Figure 4 Analyses of the spatial aggregation of recruits affected by density-dependent factors for control (left panels) and thinned plots (right panels).

The pair correlation functions g(r) from the data (black lines) are contrasted to simulation envelopes (grey area) based on maps of density-dependent covariates of each plot. For further conventions, see Fig. 2.

Comparing treatments, we found that recruits were more aggregated than expected if they were only distributed by environmental factors at less than 0.25 m in control plots (Fig. 3). In thinned plots, we found a similar pattern that of control plots but recruits were more aggregated than expected by the distribution of environmental factors at less than 0.5 m and around 2.0 m. Comparing independent plots we found that recruits were consistently more aggregated than expected by the spatial distribution of environmental factors at less than 0.5 m and around 2.0 m whereas in control plots 4 and 9 were more aggregated than expected in all distances (Appendix S2; Fig. S6).

In addition, recruits were less aggregated than expected if they were only associated with the density-dependent factors, a pattern varying over the studied years: in control plots, recruits were less aggregated than expected from 0.5 to 3.5 m in the year before thinning, and this range shrank steadily over years up to 2.2 m (Fig. 4). In thinned plots, recruits were less aggregated than expected by the density-dependent factors from 0.5 to 3.0 m; the latter pattern was consistent along the studied years (Fig. 4). Comparing independent plots, we found that such pattern was consistent over the studied years: we found that recruits were less aggregated than expected from around 0.5 to 3.0 m and beyond, despite such pattern being variable between plots (Appendix S2; Fig. S7).

According to the factors explaining the recruit’s survival, we found differences between treatments. For instance, alive recruits were in locations with higher values of the intensity function λ(x) of environmental factors, but such differences were only detected in 2008 and in control plots (Appendix S2; Fig. S8). By contrast, we found differences in the value of the intensity function λ of density-dependent factors between alive vs dead recruits in 2009 (i.e., higher values for alive recruits in control plots) and in 2010 (i.e., higher values in dead recruits in thinning plots) (Appendix S2; Fig. S8).

Discussion

Here, we aimed at studying the ecological factors conditioning the fine-scale distribution of tree recruits before and after a thinning event. We based our results on plots of small size (around 0.12 ha) which could be not valid for analyzing the spatial distribution of trees, but adequate to test (i) the influence of the proximity of trees and recruits and (ii) the relative contribution of environmental and density-dependent factors in explaining the density, aggregation and mortality of tree recruits. With the analysis of the one year before and three years after a thinning event, we have been able to discriminate between those processes related to the thinning itself from historical processes derived from the patch dynamics. We found that the aggregation pattern of recruits in thinning plots was more heterogeneous but independent of tree distance and that only translates into proportionally higher survival of recruits close to trees. By contrast, recruits in control plots were only aggregated at the local proximity, were distributed closer to trees, and had higher survival at distances beyond that they were close to trees. Our findings thus suggest that thinning favours tree-recruit facilitation, at least during the few years after thinning. Independently of the treatment, the distribution of recruits was chiefly determined by the density-dependent factors at less than 0.5 m and environmental heterogeneity at distances beyond 0.5 m, but such spatial patterns were highly variable between plots and fairly consistent over the four studied years. Despite thinning could affect the distribution of recruits, we found in our study site that patch dynamics is the main driver modulating the distribution and aggregation of tree recruits at fine scales.

Density and aggregation of trees and recruits

There is a broad evidence that thinning affects demography of forest systems, typically by boosting changes in regeneration stages (e.g., Bailey & Tappeiner, 1998; Dolanc, Gorchov & Cornejo, 2003; Yeo & Lee, 2006; Ding & Zang, 2021). Our analyses revealed differences in the within-plot density of both trees and recruits and that pattern varied between years and plots. We found that trees decreased in density along the studied years, notably in control plots, and such difference between treatments disappeared in the third year after thinning. Latter changes in tree density of control plots could be the consequence of self-thinning, a typical ecological process occurring in unmanaged forests modulating competition of trees for growing space. In our study site, P. sylvestris trees are the dominant trees and there is also strong evidence that such tree species compete for resources and affect its survival in the long-term (Primicia et al., 2013; Primicia et al., 2016; Cardil, Primicia & Castillo, 2018). We also found a steadily decreasing trend in density of recruits, notably in thinned plots, a density which arrived to comparable values to that of control plots three years after the thinning event. Thinning promotes early tree regeneration by quickly increasing the density and growth of recruits, typically stabilizing few years after thinning (e.g., Dolanc, Gorchov & Cornejo, 2003; Yeo & Lee, 2006), a pattern consistent with our results. Thus, the initial and consistently higher density of recruits in thinned plots could be a consequence of historical events likely promoting tree regeneration in thinned plots and that may suggest that recruit mapping during the periods spanning thinning events could be potentially short.

In our study site, we did not found that such comparatively higher density of recruits was substantially affected after the 2009 thinning event, which suggest strong influences of patch dynamics in our forest system. Likewise we also found that the aggregation of both trees and recruits was clearer in control than in thinned plots, which suggest that thinning re-set and homogenize patch dynamics (Oliver et al., 1996). Given that tree density did not differ between treatments after the 2009 thinning event (present work; Primicia et al., 2016), we can suggest that thinning lead to generate homogeneous environmental conditions. In addition, we observed that the density of recruits in thinning plots in 2011 did not differ from that of control plots, a pattern which suggests that patch dynamics were steadily blurring the initial higher density generated by thinning, which arrived to a similar and more homogeneous between-plot density to that of control plots.

The analysis of the spatial patterns in forest systems is key to better understand the effect of ecological processes on tree dynamics (Franklin, 2010 and references therein). In particular, tree-recruit competition (e.g., Wolf, 2005), limited dispersal of recruits from source trees (e.g., Jacquemyn et al., 2007) or environmental heterogeneity (e.g., Alcántara et al., 2018) are the most important processes structuring the aggregation of recruits. In our study site, most broad-leaf trees contribute little to seed production, meaning that recruits of broad-leaf species are mainly the consequence of a colonization process outside our study plots; F. sylvatica is an exception, with few large trees on our study plots. Despite being commonly important in most study systems (Franklin, 2010 and references therein), we assumed that the limitation of seed dispersal had low effects on the recruit aggregation, at least during our studied years. Likely, the limitation of seed dispersal of tree recruits will increase its importance as the process of vegetation succession takes place in our study system, changing from pure stands of Scots pine to mixed-forest stands.

When seed dispersal plays a minor role in explaining plant aggregation, a regular pattern gives evidence for plant-plant competition, clumping suggests that gap dynamics and favorable microsites are more important, whereas random patterning indicates prevalence of neutral processes such as random mortality. As expected, we found that recruits were aggregated at short distances, specially at less than 0.5 m and independently of the treatment. In control plots we found that recruits were randomly distributed at distances beyond 0.5 m, whereas, in thinned plots their aggregation also occurred at distances around 1.5 m and beyond. Latter trend probably derived from higher heterogeneity of environmental factors derived from the thinning event, as evidenced by other studies (e.g., Barbier, Gosselin & Balandier, 2008). In particular we found that the local aggregation around 2 m and probably derived from areas of thinned vs. un-thinned trees which generate uneven fine-scale environmental conditions associated with areas with low vs. high tree density. Overall, we found that latter spatial patterns were consistent over studied years, which suggest that the distribution and aggregation of recruits were maintained along four study years, despite that the density, notably of recruits, was steadily decreasing over this time spam (Fig. 1). Independently of the treatment, we also found strong differences in the aggregation of recruits, with positive departures at less than 1.0 m and beyond; such pattern probably derived from spatial heterogeneity within the limits of our study plots.

In resource-limited environments, adult trees influence tree recruits at their the local proximity, by competing for resources and by increasing their performance under nurse canopy (Wright, Schnitzer & Reich, 2014). Such competitive-facilitation gradient occurring between tree ontogenetic stages could be additionally modulated by between-year variability of environment, notably in water-limited climates (e.g., Andivia et al., 2018), as happens in our study site during summer season. We found that the location of recruits in thinning plots were independent to trees, whereas those in control plots were closer to trees. Despite the latter pattern being consistent throughout the studied years, our findings suggest that the tree-recruit aggregation was chiefly a consequence of environmental conditions generated after thinning. In addition we found that the local distribution of trees and recruits also depended on the patch dynamics, as we found strong differences between plots. In particular, in plots 2, 3, 5 and 9 trees and recruits were aggregated whereas in plot 4 they were repulsed from 1.5 to 6.0 m; the latter spatial patterns was probably due to the local dynamics, such as a strong tree fall as a consequence of a windstorm before the thinning event that created large openings in the local proximity of trees (Ruiz de la Cuesta et al., 2021). We additionally found that recruits in control plots had proportionally higher survival beyond 2.0 m from trees, probably as a consequence of higher tree-recruit competence for resources; for instance, rainfall was proportionally low during the summer season in 2008 and that potentially affected the tree-recruit competence in that year. Compared to control plots, our results led to suggest that recruits in thinning plots were favored by tree-recruit facilitation, probably as a consequence of proportionally lower aggregation of recruits in the local proximity. Likewise, we found that distance-dependent survival of recruits to trees was clearer in control plots and was highly variable between plots and that pattern probably derived from environmental heterogeneity in our study site.

Factors explaining the density and aggregation of recruits

Plant communities are typically distributed in non-random spatial patterns as a consequence of underlying ecological processes at each locality, which include environmental filtering, plant-plant interactions and neutral processes (Wiegand & Moloney, 2014). Traditionally, it has been hypothesized that biotic interactions shape the spatial density and aggregation of plant communities at fine scales as a result of the proximity to immediate neighbors, whereas environment filters plant density and aggregation at medium-to-larger scales as a consequence of environmental gradients (Götzenberger et al., 2012 and references therein). However, recent studies suggest that fine-scale distribution of environmental heterogeneity could likewise affect at the same spatial scales than biotic interactions (Perry, Enright & Lamont, 2008; Shen et al., 2013; Pescador et al., 2020). In our case we expected that the spatial density and aggregation of recruits were derived from a combination of environmental and density-dependent factors, likely interacting with the treatment.

Independently of the treatment, we found that environmental processes chiefly influenced the aggregation of recruits at distances between 0.25 m and 1.5 m, and were determined by positive effects of canopy richness. Our study site is composed by mosaics of pure canopies of P. sylvestris and mixed canopies dominated by P. sylvestris and F. sylvatica, and small amounts of areas composed by broad-leaf species (Appendix S1; Fig. S3). In our study system, canopies of mixed patches favor rainfall interception and dry deposition of nutrients (Primicia, 2012), suggesting that richer canopies lead to increase the availability or resources in the soil patches beneath them. In addition and in the same study system, Yeste et al. (2021) found lower C/N ratio in the soil close to beech stems, suggesting better quality in mixed litter in comparison to pine litter. We additionally found that recruits were associated with higher amounts of the common bracken P. aquilinum, a typical forest floor fern which produces dense stands and lead to the stagnation of forest succession by chemically inhibiting seedling growth (e.g., Humphrey & Swaine, 1997). In our study site we generally found a positive association between recruit density and the proportion of P. aquilinum (Appendix S2; Table S1), but such effects varied between plots both in positive and negative effects (Appendix S2; Table S3). In addition, we also found that the proportion of the bramble R. ulmifolius affected negatively on the density of recruits in two plots probably as a consequence of competition for light in forest openings (e.g., Arellano-Cataldo & Smith-Ramírez, 2016). Latter species have the capacity to invade rapidly new areas, suggesting that the variation of intensity and direction of both species on recruit aggregation is probably a consequence of the local invasion stage.

Besides all above spatial patterns, we additionally found strong differences between plots, which were consistent over the studied years. In control plots, for instance, canopy richness affected positively on recruit density in plots 3 and 9, except in plot 4 probably as a consequence of particular patch dynamics (Appendix S2; Table S3) generating strong differences in canopy. Likewise, plots 3 and 4, were proportionally more affected by light variability whereas plots 2, 7 and 9 by forest understory. Taking into account the plot location within the study site (Appendix S1; Fig. S1), a west-to-east gradient was likely influencing light variability and forest understory, respectively, and promoted changes in the local density and aggregation of recruits. Thus, the differences between plots could likely be a consequence of filtering derived from environmental gradients occurring at distances beyond the plot’s limits (>40 m).

Biotic interactions between recruits are one of the most important drivers structuring spatial patterns of plants (Callaway & Walker, 1997), influencing the distribution and aggregation of tree individuals up to few meters (Wiegand et al., 2009; Espinosa et al., 2016). In our case density-dependent factors mainly affected the aggregation of recruits at less that 0.5 m. Specifically, recruits were positively aggregated in areas with larger recruits, and, in the case of control plots, in areas with higher density of trees; latter finding could be derived from expansion of tree crowns of broad-leaf trees after thinning, but the effect of other ecological processes cannot be ruled out. We additionally found differences between plots, but latter pattern was consistent along years (Appendix S2; Table S4). In control plots, for instance, we found aggregation of recruits to trees in plots 3 and 9, whereas in plot 4 we found repulsion at around 0.25 m probably derived from strong differences in forest canopy (Appendix S2; Fig. S7). Finally, we found that recruits were denser in areas richer in recruit species and that pattern was consistent between plots and independent of the treatment. Latter pattern suggests that neutral processes and/or facilitation likely operates at tree regeneration and those processes persisted from one tree generation into subsequent ones, probably associated with regeneration hotspots with positive-feedback processes (Hampe et al., 2008; Batllori et al., 2009). Further studies aimed at identifying and associate canopy and recruits at fine scales would shed new light on the importance of biotic interactions on assembling forest communities.

Tree recruits are able to quickly respond to changes in environmental and density-dependent processes (Moeur, 1997; Kenkel, 1988; Plotkin, Chave & Ashton, 2002), and their survival has strong implications on regeneration of forest systems. In our case we found that alive recruits were found in areas with better environmental conditions, whereas we did not find differences in the conditions of density-dependent factors accounting for the location of alive or dead recruits. Thinning typically reduces the density of trees and alleviates the tree-recruit competition, but, at the same time, increases recruit competition in the long term (Wild et al., 2014). When taking into account environmental factors, we found less differences (despite being significant) in the local conditions of alive vs dead recruits in thinned plots, compared to control plots. Thus our results suggest that thinning itself does not significantly affect recruit’s survival, but its effect alleviates recruit competition for resources as a consequence of the disturbance itself and environmental homogenization generated after thinning.

Conclusion

Disturbances in forests modify environmental context and alleviate the competition of neighbouring trees by reducing their density and aggregation (Gratzer et al., 2004). In our case, we found that thinning affects on tree regeneration, particularly (a) by alleviating tree-recruit competition and (b) by modifying the spatial heterogeneity of the environment and the distribution of recruits in each plot. Independently of the treatment, we found strong differences between plots, leading to suggest that differences derived from the local dynamics and historical processes. In addition, we were not able to detect competition, neither between recruits nor trees, which suggests that tree recruitment of our study system was modulated by neutral and/or positive biotic interactions, at least during the three years after the thinning event. Regeneration stages are highly sensitive to changes in environmental conditions (Kitajima & Fenner, 2000) as a consequence of the fine-scale distribution of environment resources and to contagious biotic processes related to dispersal limitation and competition (Schupp, 1995; Wright, 2002). We found that the density and aggregation of recruits were not directly related to the thinning event, whereas the recruit’s survival depended more on the within-plot environmental variability, derived from environmental filtering generated by the thinning itself. At least in our study system, thinning provides new opportunities for forest regeneration and environmental heterogeneity, boosting the local forest dynamics. Our study showed how detailed analyses of spatial patterns of recruits can help to decompose the relative contribution of ecological processes structuring plant communities and at which scales such processes operate (Wiegand & Moloney, 2014).

Supplemental Information

Supplemental Information 1 Detailed information of material and methods

Click here for additional data file.

Supplemental Information 2 Detailed information of the results of Point-pattern analyses for plots and treatments

Click here for additional data file.

We thank Manaik Rivière, Iosu Ganchegui and Amaia Berastegui for their help during some field surveys in 2010 and 2011.

Additional Information and Declarations

Competing Interests

Author Contributions

Field Study Permissions

Data Availability

The authors declare there are no competing interests.

Javier Rodríguez-Pérez analyzed the data, prepared figures and/or tables, authored or reviewed drafts of the article, and approved the final draft.

Bosco Imbert conceived and designed the experiments, performed the experiments, authored or reviewed drafts of the article, and approved the final draft.

Javier Peralta performed the experiments, prepared figures and/or tables, authored or reviewed drafts of the article, and approved the final draft.

The following information was supplied relating to field study approvals (i.e., approving body and any reference numbers):

The Government of Navarra, council of Aspurz approved this study.

The following information was supplied regarding data availability:

Raw data is available at Zenodo:

Bosco Imbert. (2022). Environment and density-dependency explain the fine-scale distribution of tree recruits before and after thinning [Data set]. Zenodo. https://doi.org/10.5281/zenodo.5845114.

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
