# Peer review of "Environment and density-dependency explain the fine-scale aggregation of tree recruits before and after thinning in a mixed forest of Southern Europe"

_PeerJ, doi:10.7717/peerj.13892_

## Round 0.1 · original submission · Major Revisions

The manuscript presents an interesting scientific study. However, there are also some issues, mostly related to the Methods section and following the conclusions presented in the Discussion section.

Reviewer 1 ·

Basic reporting

Using data collected during four years, one year before and three years after a thinning event, the authors studied the distribution of recruited trees in mapped plots. The overall research work is interesting, although the document is susceptible to improvement. In particular, there is a lack of general editing, perhaps the authors should consider compacting the information as much as possible throughout the document. Particularly in the Methods and Discussion sections.
The use of digression should be used only when it is really necessary. Rather, the authors should strive to express the ideas as clearly as possible in the main text that drives the descriptive discourse, without overusing additional clarifications.

Experimental design

1. From my reading subjectivity, the Methods section is very “heavy”, sometimes confusing. For example, << In the mid-1960’s, a strip clear-cutting or forest harvest of all trees was performed in the study’s area. However, later on, only Scots pine trees have been thinned following the silvicultural trends applied currently in Navarre on mixed stands with European beech as a secondary species (Primicia et al., 2013). Consequently, since the beginning of our studies in 1999, we have shown that beech cover is expanding in our stand (Primicia et al., 2016). Indeed by 2017 beech litterfall already exceeded that of pine (Gonzalez de Andres et al. 2019).>> lines 126-131, I am not sure if these paragraphs contribute significantly to a better understanding of the study. I would think not. If it serves a function for the audience, it will have to be rewritten.
.
3. I think most of the paragraphs could be reduced by placing only the necessary information, in a direct and condensed manner. In such a way that the audience can easily follow the research work. The need the authors feel to make additional clarifications by using "i.e." in parentheses, is precisely because they do not ground an idea in a concrete and concise way.

4. In Methods section after the description of the study area (L107-137), The descriptive discourse will be more fluent and clearer if the authors list the steps followed to obtain the results. Perhaps using a flowchart or a table.

5. In lines 178, 181, the authors write in parentheses "see below". It would be better to refer to a table or a figure where the audience can immediately compare the information referred to.

6. The authors should see if it is feasible and pertinent to incorporate a table in the Methods section. One way to reduce the text would be to incorporate a table showing the information recorded in the field for different years in a summarized form. Even include a summary of the descriptive statistics of the quantitative information collected for each year.

7. L. 183. The authors write “For details about methodology, see Appendix S1”. However, Appendix S1, although very valuable, contains only supplementary information. I suggest that it be deleted or reworded. The information that is as detailed and timely as possible should be in the main document.
8. In the Methods section, or at least, in the subchapter “Summary statistics of spatial-point patterns “
I suggest that you list the steps followed. What was done first? What was done next? As you list the steps, you can mention the functions used, clarifying what for. In such a way that the audience can follow the steps. IMPORTANT: These steps should correspond to the order in which the results are presented.

9. L. 240-256. When describing the formulas, I think it is good to describe what they are, but perhaps it is more important to put, what it was used for in the study or for what purpose it was used. In a concrete way without rambling.

10. L. 267 “spatstat”, is an R package. Please clarify this.

11. Again, L. 278, 315, authors should consider the use of bullets or the use of numbers to list the steps, when describing the Analysis 1 (L.278) & Analysis 2 (L.315), in a way that the audience can follow the whole process of obtaining the results in a timely manner. In these steps, the notations of the equations used could be incorporated.

Validity of the findings

1. In the results section, there is an excessive use of "i.e.", in some parts, when making an additional clarification in parentheses. For example, L. 405-407 <<…Specifically, the canopy richness (i.e. recruits were denser in richer tree canopies), the treatment (i.e. denser in thinned plots), and the proportion of S. purum (i.e. denser in areas abundant in S. purum) were the environmental…>> Perhaps use in the first paragraphs, and then eliminate them, since their excessive use avoids a fluent reading. This reiterative use of "i.e" is more reiterative in chapter “Factors explaining the density and aggregation of recruits”, L. 404, 421. I suggest that these clarifications be eliminated or incorporated into the main text.

2. L.587 remove “ i.e.”, and leave only “ > 40 m”.

3. In L. 594-595 “We additionally found strong differences between plots”, cite a table or figure that confirms "the strong difference". Preferably numerical evidence from a hypothesis test.

4. L. 598. Replace "see above" with a citation of figure, appendix or table.

5. In Appendix S1, Figure S1. Which one is "a" and which is "b"? The figures do not have letters.

6. In Supporting Information, Figure S2, I suggest that the colors of the circles used as captions match the colors of the images shown in the main fig, instead of black. In addition, the letters of the captions overlap with the ordinate axis. Overlapping should be avoided.

7. Instead write “i.e. 3, 4 and 9 plots, respectively, from left to right”, better to cite the numbers of figures.

8. In the Discussion section, authors should consider including two or three lines emphasizing the possible use or application of their findings. For example, at the end of the discussion.

Additional comments

It is an interesting paper potentially publishable in PeerJ. The method used is in accordance with the research objectives.

Reviewer 2 ·

Basic reporting

The study "Environment and density-dependency explain the fine-scale distribution of tree recruits before and after thinning" by Javier Rodríguez-Pérez is a very interesting study which suggests that thinning affect the distribution of newly recruits individuals in mixed forest at Navarra, Spain. The authors have an extensive literature review and an adequate study design and statistical analysis.

I only would suggest that the tile should include reference to the type of forest and the region (country) where the authors conducted the study. Similarly, the abstract should include some of this information too.

Experimental design

The experimental design is adequate and is within the aims and scope of Peer Journal.
I only suggest a map indicating where the study area is located.

Validity of the findings

The findings are clearly stated and supported by robust and adequated statistically analysis.

Additional comments

In the four figures, I suggest the authors describe in the legends the meaning of the x variable (Distance r (m)).

Reviewer 3 ·

Basic reporting

The manuscript presents an interesting scientific study that studies the influence of forest thinning on the spatial distribution of tree individuals with a strong focus on recruits. The setup with fully-mapped thinned and control study plots in spatial proximity and thus assumed environmental homogeneity is very interesting. Furthermore, spatial point pattern analyses are a powerful tool to reveal underlying ecological processes and are well suited for the research objectives. The manuscript is well written and overall good to understand, however, it would benefit from some minor language editing. More important, there are also some issues, mostly related to the Methods section and following the conclusions presented in the Discussion section.

Experimental design

The manuscript would benefit from a stronger connection of null models to underlying ecological processes. This is the most crucial, but also most difficult part of spatial point pattern analysis. Related to this, the null model for the bivariate analysis of recruits and trees is not well suited (please see comment in the specific remarks section for further details). Furthermore, the use of environmental variables and intensity maps as proxies for environmental conditions is not always clear and the manuscript would benefit from a more precise explanation. Especially since it is not always clear what is cause and effect when intensity maps that are based on the point pattern are used to explain the point pattern.

Even though the setup is unique and interesting the time period between thinning events and recruits mapping is very short. Related to this, the study plots are rather small in size. While this is not necessarily an issue per se, it should be at least discussed in the Discussion section and how this might influence the results of the research study.

Validity of the findings

The applied null model for the bivariate analysis (“random labelling”) is not perfectly suited and thus interpretations of the results are less meaningful (please see comment in the specific remarks section for further details).

One major ecological process responsible for many aggregated tree patterns around the world, namely limited seed dispersal, is hardly mentioned and discussed in the manuscript at all. Also, even though the study plots are assumed to be homogenous, clustering at more favorable growing conditions (and thus heterogeneous soil conditions) is another very common process for clustered patterns. While this is touched on here and there, it could be discussed more prominent.

Additional comments

Lines 47: What are the differences between distribution and aggregation? Aggregation is just one specific distribution.
Lines 59/61: The transition between the paragraphs could be improved by explaining that thinning is one specific disturbance.
Lines 91: What is meant by “relationship between individuals and species”?
Lines 93: This is slightly contradicting Lines 75 in which it is stated that variability might interact (with which I agree).
Lines 96: Especially for the first research objective, the line of reasoning is not completely clear.
Lines 98: What about heterogeneous environmental conditions as a reason for clustering?
Lines 143: Please reword in a way that two different thinning intensities were applied (20% and 30%) and one control treatment (no thinning, 0%).
Line 159: How are recruits defined? Was there a size threshold?
Line 168: Were recruits that grew > 200 cm automatically included as trees?
Lines 203: How was the difference between forest understory and recruits defined?
Lines 226: While using the simulation envelopes in comparison with the observed summary function is a measure of significance that is often applied, it might suffer from a Type I error inflation. Thus, additional Goodness-of-Fit tests are strongly suggested (see Baddeley et al., 2014; Loosmore and Ford, 2006).
Lines 251: Regular patterns are indicated by a g(r)<1.
Lines 253: Why was the J-function used? What were the advantages of this function vs. e.g., the pair-correlation function?
Lines: 267: Why were the analyses restricted to regular shaped areas? Most edge corrections can deal with irregular-shaped areas (to a degree).
Lines 289: Random labeling is the wrong null model for the research questions (Goreaud and Pellissier, 2003). Independence or even better antecedent conditions (Wiegand and Moloney, 2004) would be a better null model. The hypothesis assumes a priori processes to shape the different patterns. This means that the assumed process that shaped the tree pattern is a different assumed process that shaped the recruit pattern. Random labeling assumes that the processes that shaped the location of the points are the same and not really of interest but focuses on a a posteriori process that acts on the established pattern and leads to different states of the individuals (e.g., dead vs. alive).
Lines 284/299: What was the reason for different edge corrections across the methods?
Lines 326: Was there a correlation between the 12 different maps? This might be the case, especially for all light-related variables.
Lines 344: The difference between (i) and (ii) is not completely clear. Some are measured directly, some are local intensity maps λ (x,y) based on the point pattern, which is then used to explain the point pattern again.
Lines 353: Related to the previous point, cause and effect are not completely clear.
Lines 365: The decrease described in Fig. 1b is less prominent than the decrease described in Fig. 1a.
Lines 367: Looking at Fig. 1c, there seems to be a difference between treatments.
Lines 377: Adding letters to the panels would make the figure easier to read (e.g., Fig. 2a, 2b, 2c, …).
Lines 378: Small deviations from the simulation envelopes should not be over-interpreted (see e.g. Wiegand and Moloney, 2014).
Lines 412: Again, cause and effect are not completely clear.
Lines 422: In this case, a heterogenous Thomas process might be a more suitable null model. Limited seed dispersal might be another process leading to aggregated patterns.
Lines 451: Here and throughout the manuscript the usage of density-dependent factors and tree-recruit and recruit-recruit processes is slightly confusing. What is the difference?
Lines 459: Only recruit survival was used as a mark, but not recruit size. This might give more insights into facilitation effects (see e.g. Schleicher et al., 2011).
Lines 462: Limited seed dispersal might be another process leading to aggregated patterns.
Lines 486: Maybe the time periods between thinning events and recruits mapping were too short.
Lines 502: Limited seed dispersal is another prominent process that can lead to clumping.

---

## Round 0.2 · Minor Revisions

The manuscript has been significantly improved compared to an earlier version. However, Rev. 3 requires some improvements.

Reviewer 1 ·

Basic reporting

No comment.

Experimental design

I think that by moving the subchapter "Summary statistics of spatial point patterns" in Appendix S1 and including a new table summarizing the analyses, helps to better understand the steps followed in the data analysis.

Validity of the findings

Errors in the description of the figures have been corrected. The figures were also improved, especially the supplementary material.

Additional comments

The structure of the current document has been improved, achieving greater fluidity than the first version.
The inclusion of the "Summary statistics of spatial-point patterns" subchapter in Appendix S1 and the inclusion of a new table summarizing the analyses contribute to a better understanding of the steps followed in the data analysis.

In general, most of the suggestions I made were attended by the authors and I have no objection to this paper being published in the PeerJ Journal.

Reviewer 3 ·

Basic reporting

The manuscript improved immensely from an earlier version and especially the applied null models and their interpretation are now correct and appropriate. Overall, the authors did a very good job addressing most to all issues raised about an earlier version of the manuscript.

Experimental design

First, even though suggested by another reviewer, I think at least one or two sentences describing what the summary functions do would be beneficial for many readers in order to understand the results and their interpretation.
Related to this, while not being “wrong”, it is unclear why for most analyses the pair-correlation function g(r) was used (non-cumulative, imho superior), but once the L(r)-function was used (cumulative). Also, it is still unclear why the analyses were restricted to rectangular areas since the now applied edge correction works also for non- rectangular study plots. But, these are not major issues per se, just points not completely clear.

Validity of the findings

Last, it is now addresses at several points within the manuscript why seed dispersal is not an important processes for the spatial point patterns. However, I do not follow the given reasoning because also the currently present recruitments had to arrive as seeds/seedlings in the study area somehow, even if mature source trees are outside (the small) study plots.

Additional comments

Line 495: Please make sure that this sentence is correct as written (“In thinning plots we found that […]”).

---

## Round 0.3 · accepted · Accept

All requested modifications of the reviewers was enforced. However, I have 3 more comments to improve the manuscript:

F. sylvativa to F. sylvatica,

n.s. p > 0.05 to n.s. p >= 0.05,

and define what is "c.".

Reviewer 3 ·

Basic reporting

The authors did a great job addressing the remaining issues and I appreciate their effort going through three rounds of revisions. I think the manuscript improved by this and is now an even more interesting contribution to the field. I do not have any other objections and suggest to accept and publish the manuscript.

Experimental design

See comment 1. Basic reporting

Validity of the findings

See comment 1. Basic reporting

Additional comments

See comment 1. Basic reporting